# CRISPRi is not strand-specific at all loci and redefines the transcriptional landscape

Françoise S Howe[1], Andrew Russell[1], Anna R Lamstaes[1], Afaf El-Sagheer[2,3], Anitha Nair[1], Tom Brown[2], Jane Mellor[1]*

[1]Department of Biochemistry, University of Oxford, Oxford, United Kingdom; [2]Department of Chemistry, Chemistry Research Laboratory, University of Oxford, Oxford, United Kingdom; [3]Chemistry Branch, Faculty of Petroleum and Mining Engineering, Suez University, Suez, Egypt

**Abstract** CRISPRi, an adapted CRISPR-Cas9 system, is proposed to act as a strand-specific roadblock to repress transcription in eukaryotic cells using guide RNAs (sgRNAs) to target catalytically inactive Cas9 (dCas9) and offers an alternative to genetic interventions for studying pervasive antisense transcription. Here, we successfully use click chemistry to construct DNA templates for sgRNA expression and show, rather than acting simply as a roadblock, sgRNA/dCas9 binding creates an environment that is permissive for transcription initiation/termination, thus generating novel sense and antisense transcripts. At *HMS2* in *Saccharomyces cerevisiae*, sgRNA/dCas9 targeting to the non-template strand for antisense transcription results in antisense transcription termination, premature termination of a proportion of sense transcripts and initiation of a novel antisense transcript downstream of the sgRNA/dCas9-binding site. This redefinition of the transcriptional landscape by CRISPRi demonstrates that it is not strand-specific and highlights the controls and locus understanding required to properly interpret results from CRISPRi interventions.
DOI: https://doi.org/10.7554/eLife.29878.001

*For correspondence:
jane.mellor@bioch.ox.ac.uk

## Introduction

Eukaryotic genomes are pervasively transcribed but the effect of this inter- and intragenic transcription is not fully understood (*Mellor et al., 2016*). Of particular interest is the function of antisense transcription, which alters the chromatin in the vicinity of sense promoters (*Lavender et al., 2016*; *Murray et al., 2015*; *Pelechano and Steinmetz, 2013*), and is associated with repression, activation or no change in levels of the corresponding sense transcript (*Murray et al., 2015*). Discerning the mechanism by which antisense transcription functions in gene regulation is confounded by the difficulty in ablating antisense transcription without direct genetic intervention (*Bassett et al., 2014*). A new approach, CRISPR interference (CRISPRi) (*Qi et al., 2013*), can circumvent these issues by avoiding the need to manipulate endogenous DNA sequences. An endonucleolytically dead version of Cas9 (dCas9) is recruited by single base-pairing guide RNAs (sgRNAs) targeting the non-template (NT) DNA strand, where it blocks transcription strand-specifically at the loci tested (*Lenstra et al., 2015*; *Qi et al., 2013*). In a further adaptation, dCas9 fusion with transcriptional repressors or activators allows both negative and positive regulations of transcription, respectively (*Gilbert et al., 2013*; *Gilbert et al., 2014*), but the strand-specificity is lost and thus is not suitable for strand-specific repression of antisense transcription. The sgRNA compenent of the CRISPRi system consists of two regions: a constant region (82 nt) that binds to dCas9 and a variable region (20 nt) that is responsible for targeting. The modular nature of the sgRNA lends itself to production from DNA templates

constructed using synthetic copper(I)-catalysed alkyne-azide cycloaddition (CuAAC) click chemistry (*El-Sagheer and Brown, 2010*). The stable artificial triazole DNA linker generated is biocompatibile, being read and accurately copied by DNA and RNA polymerases (*Birts et al., 2014*; *El-Sagheer and Brown, 2011*; *El-Sagheer et al., 2011*). We show that click chemistry is an efficient method for sgRNA template construction and that when combined with dCas9, these sgRNAs are as effective as sgRNAs produced from chemically synthesised full-length DNA templates at reducing levels of transcripts in vivo. However, it is still not completely understood how the CRISPRi system functions strand-specifically or why sgRNAs must target the non-template strand.

Here, we use the CRISPRi system to study the effect of blocking antisense transcription at loci with well-characterised sense:antisense transcript pairs. We have previously used a promoter deletion of the antisense transcript *SUT650* at the *HMS2* locus to show that *SUT650* represses *HMS2* sense transcription (*Nguyen et al., 2014*). Now we use CRISPRi to examine the effects of blocking *SUT650* antisense transcription to ask (i) whether *SUT650* represses *HMS2* sense transcription *without* a genetic intervention and (ii) whether CRISPRi is strand-specific using, in addition to *HMS2*, an engineered *GAL1* gene with a well-characterised antisense transcript (*Murray et al., 2012*, *2015*).

The main conclusion from this study is that CRISPRi at the *HMS2* locus is not fully strand-specific and results in (i) premature termination of the sense transcript and (ii) initiation of a new unstable antisense transcript in the vicinity of the sgRNA binding site. As transcription from this new antisense initiation site extends in to the *HMS2* promoter, there is no net change in *HMS2* sense transcript levels. Thus, CRISPRi redefines the transcriptional landscape at *HMS2*. This suggests that routine use of CRISPRi for gene expression analysis will require rigorous analysis of transcript integrity and function before conclusions can be drawn.

## Results and discussion

### DNA templates for sgRNA production can be made using click chemistry

CRISPRi-mediated transcriptional repression requires co-expression of a mature sgRNA and dCas9 (*Figure 1A*). A small library of single-stranded DNAs, comprised of templates for sgRNA variable regions, were joined to the constant region using click chemistry (*Figure 1B*) and used successfully as templates for PCR, with no significantly different efficiencies when compared to control full-length synthesised oligonucleotides (*Figure 1C*). The PCR products were inserted in place of a *URA3* selection cassette in the endogenous *snR52* locus for expression of a transcript that is then processed to form the mature nuclear-retained sgRNA (*Figure 1A*). Levels of dCas9 protein were uniform between strains (*Figure 1—figure supplement 1*). Neither insertion of *URA3* into snR52 nor dCas9 expression in the control strains affected growth rate, although strains expressing some sgRNAs grew more slowly indicating a physiological effect (*Figure 1D*).

### CRISPRi represses the production of antisense transcripts at *HMS2* and *GAL1*

CRISPRi represses transcription when sgRNAs/dCas9 are targeted to the non-template (NT) strand next to a protospacer adjacent motif (PAM) (*Qi et al., 2013*) (*Figure 2A*). Firstly, we used an engineered version of *GAL1* that has a stable antisense transcript (*GAL1* AS) initiating within an *ADH1* terminator inserted into the *GAL1* coding region (*Murray et al., 2012*) (*Figure 2B*). *GAL1* AS is present in cells grown in glucose-containing media when the *GAL1* gene is repressed and is reduced as cells are switched into galactose-containing media and *GAL1* sense is induced (*Figure 2C* lanes 1–3). We designed sgRNAs adjacent to two PAM sequences on the non-template strand near the antisense transcription start site (TSS) (AS+28NT and AS+112NT) and a third strand-specificity control sgRNA on the template strand in this region (AS + 93T) (*Figure 2B*). Only sgRNA AS+112NT/dCas9 caused significant (p=0.004) reduction in *GAL1* antisense transcript levels, as assessed by Northern blotting (*Figure 2C,D*).

Next, we examined the *HMS2* locus which has a stable antisense transcript, *SUT650*, initiating within the 3' coding region of *HMS2* (*Nguyen et al., 2014*; *Pelechano et al., 2013*). We designed three *SUT650*-targeting sgRNAs, located 59, 243 and 276 nucleotides downstream of the major *SUT650* TSS on the non-template strand (sgRNAs AS+59NT, AS+243NT, AS+276NT, respectively)

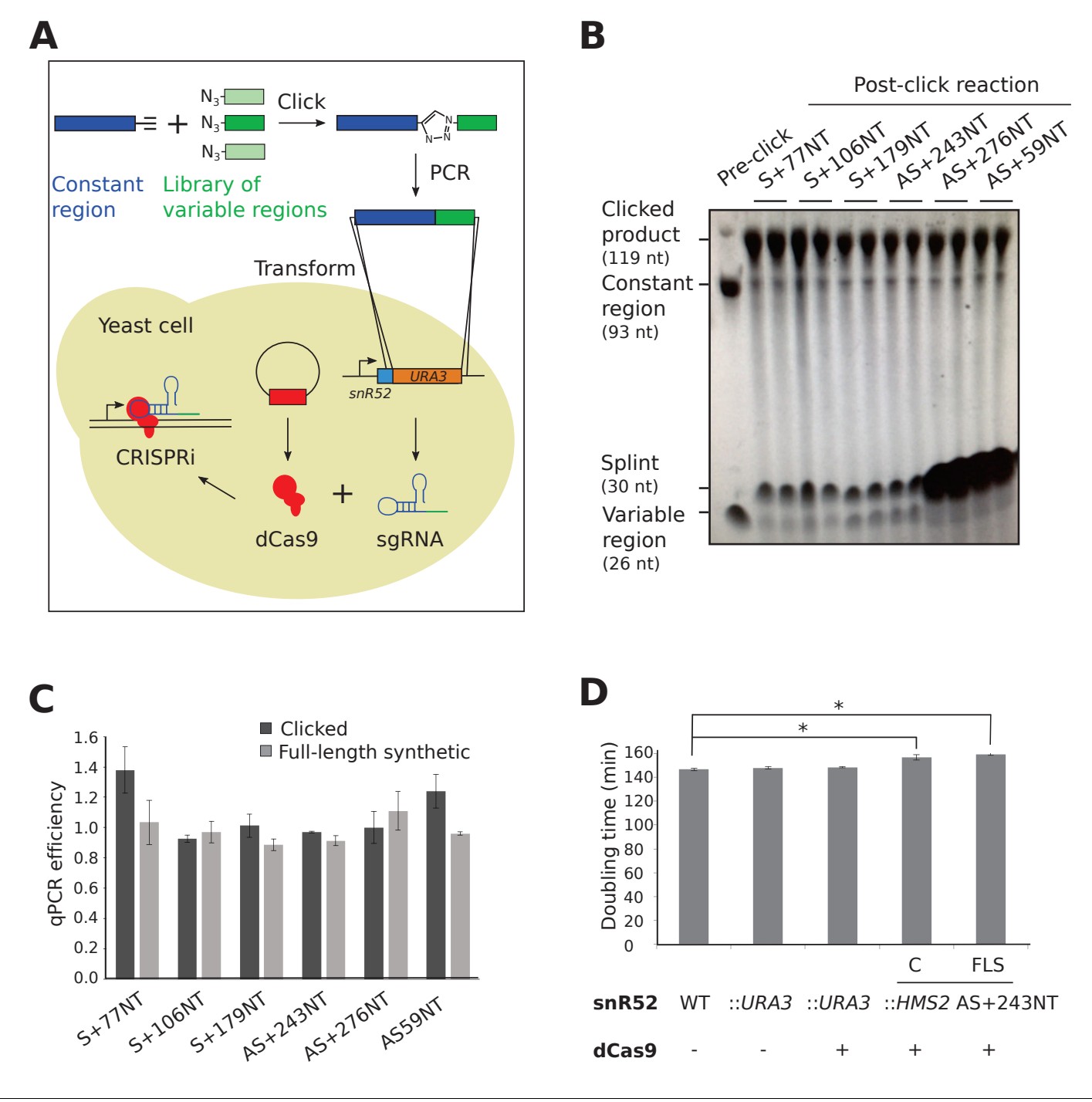

**Figure 1.** Experimental strategy for sgRNA template synthesis and incorporation into yeast. (**A**) An outline of the experiment. DNA templates for sgRNA production were generated using a click chemistry reaction between a single-stranded DNA oligonucleotide with a 3' alkyne group encoding the constant region (dark blue) and a number of single-stranded DNA oligonucleotides with 5' azido groups encoding the different variable regions (green). The resulting single-stranded DNA oligonucleotide was purified, amplified by PCR and transformed into yeast to replace the *URA3* selection cassette that had previously been introduced into the endogenous *snR52* locus. Correct insertion was confirmed by Sanger sequencing. The expression of the sgRNA is driven from the endogenous RNA polymerase III *snR52* promoter and processed to produce a mature sgRNA. The sgRNA couples with enzymatically dead Cas9 (dCas9, red), expressed under the control of the *TDH3* promoter off a plasmid, to block transcription. See Materials and methods for more detail. (**B**) A polyacrylamide gel visualised by UV shadowing reveals a high efficiency of the click reaction for all the *HMS2* constructs (see *Figure 2E* and *Figure 3—figure supplement 1* for positions of the sgRNA binding sites). The sizes of the variable region, DNA splint, constant region and clicked product are indicated. (**C**) PCR efficiencies of the clicked and full-length synthetic DNA oligonucleotides as

*Figure 1 continued on next page*

*Figure 1 continued*

measured by qPCR with a serial dilution series. N = 3, errors are standard error of the mean (SEM), all differences are not significant p>0.05. See *Source data 1*, tab 2. (D) Doubling times (min) of the indicated yeast strains grown in complete synthetic media minus leucine. C, strains constructed using clicked oligonucleotides; FLS, strains constructed using full-length synthetic oligonucleotides. N = 10, errors are SEM, *p<0.05. See *Source data 1*, tabs 3–4.

DOI: https://doi.org/10.7554/eLife.29878.002

The following figure supplement is available for figure 1:

**Figure supplement 1.** Levels of dCas9 protein in the experimental strains.

DOI: https://doi.org/10.7554/eLife.29878.003

(*Figure 2E*). Note that there is considerable heterogeneity in the *SUT650* TSS (*Figure 2—figure supplement 1*) and thus in some cells, sgRNA AS+59NT may be binding upstream of the *SUT650* TSS. AS+243NT/dCas9 significantly (p=0.011) reduced *SUT650* (*Figure 2F* lanes 1 and 3, 2G) and this repression was comparable for the clicked and control full-length synthetic constructs (*Figure 2G*). However, there was no obvious effect of the other two sgRNAs/dCas9 on *SUT650* levels (*Figure 2F* lanes 1, 4 and 5, 2G). Consistently, we could only detect dCas9 binding to the site of sgRNA AS +243NT binding and not to the other two *SUT650*-targeting sgRNA binding sites tested (*Figure 2—figure supplement 2*). CRISPR efficiency has been linked to nucleosome occupancy (*Horlbeck et al., 2016b*; *Isaac et al., 2016*), but we found no correlation between the level of repression and the nucleosome occupancy (*Knight et al., 2014*) at each of the *SUT650* sgRNA target regions (*Figure 2—figure supplement 3*). However, other factors such as sequence determinants can also influence repression (*Horlbeck et al., 2016a*). These results at *GAL1* and *HMS2* confirm CRISPRi is suitable for reducing levels of antisense transcripts in yeast but controls are needed for each sgRNA designed to ensure that repression has been achieved.

## CRISPRi repression of *SUT650* induces a new shorter antisense transcript at *HMS2*

At *HMS2*, sgRNA AS+243NT/dCas9 was able to reduce *SUT650* transcript levels significantly (*Figure 2F,G*). Since *SUT650* is a substrate for the major cytoplasmic 5′−3′ exonuclease Xrn1 (*Figure 2F*, lanes 1 and 2), we investigated whether the CRISPRi-induced *SUT650* reduction was due to direct repression of *SUT650* transcription and/or a reduction in *SUT650* stability. Whilst *SUT650* was not stabilised upon *XRN1* deletion in the strain expressing AS+243NT/dCas9, supporting previous studies that CRISPRi operates via a transcriptional block (*Qi et al., 2013*) (*Figure 2F*, lanes 3 and 6, transcript B), we detected a shorter antisense transcript (*Figure 2F*, lane 6, transcript B[t]). To map transcript B[t], we used a series of Northern blotting probes across the locus (*Figure 2H*). With antisense-specific probes H1 and H2 in the *HMS2* gene promoter and 5′ coding region, respectively, we could detect antisense transcript B[t], indicating that it extends into the *HMS2* sense promoter. However, probe H4 at the 3′ end of the *HMS2* gene was unable to hybridise to B[t], indicating that B[t] initiates upstream of this site. So whilst CRISPRi successfully blocked the original *SUT650*, a new Xrn1-sensitive unstable antisense transcript (B[t]) was created. This is potentially due to changes in the chromatin structure caused by AS+243NT/dCas9 binding to this region (*Horlbeck et al., 2016b*; *Isaac et al., 2016*) or by blocking *SUT650*, creating an environment that is permissive for transcription initiation (*Murray et al., 2012*). We note that using probe H4 we were also unable to detect *SUT650* initiating from its WT start site but terminating at the AS+243 NT binding site, due to its small size (~243 nt).

## CRISPRi-induced *GAL1* AS repression does not affect the *GAL1* sense transcript

We tested whether *GAL1* AS repression by AS+112NT/dCas9 affected the *GAL1* sense transcript. Previously we mutated a TATA-like sequence to ablate antisense transcription but observed no difference in *GAL1* sense transcript levels in the population under repressive or activating conditions (*Murray et al., 2015*). Using CRISPRi, we also observed no significant change in *GAL1* sense transcript levels or size (*Figure 3A,B*). Additionally, we observed no leaky expression of the *GAL1* sense transcript under repressive conditions (glucose) when the *GAL1* AS transcript was blocked

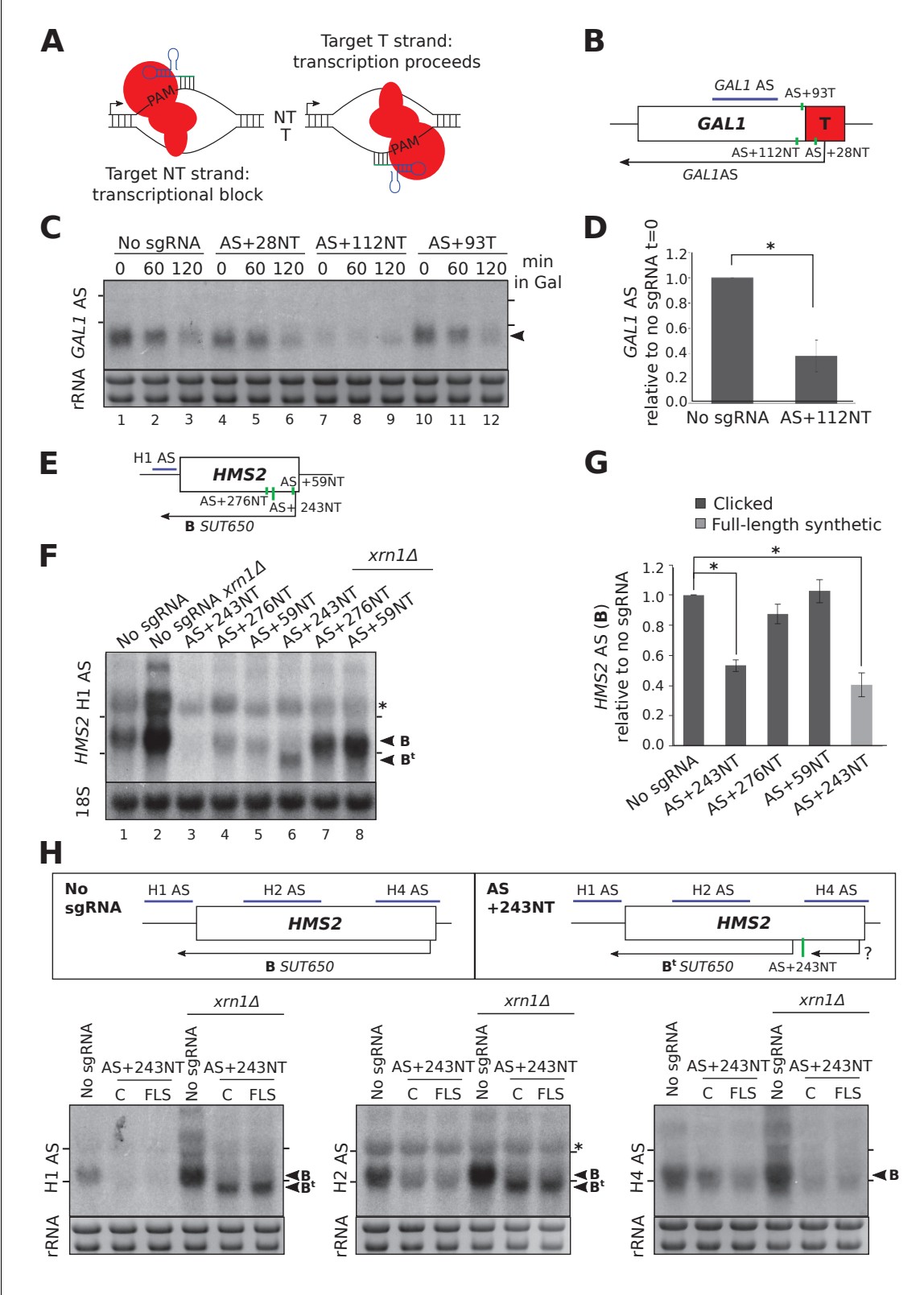

**Figure 2.** CRISPRi reduces antisense transcripts at *GAL1* and *HMS2*. (**A**) A schematic demonstrating the previously reported requirement of non-template strand targeting by the sgRNA/dCas9 complex for strand-specific transcriptional repression. Arrows indicate the direction of transcription. NT, non-template; T, template; red, dCas9; PAM, protospacer adjacent motif; green/blue, sgRNA. (**B**) A schematic of the engineered *GAL1* locus showing the site of insertion of the *ADH1* terminator (T, red box), the transcription start site for the stable *GAL1* antisense transcript and the positions of the

*Figure 2 continued on next page*

*Figure 2 continued*

designed sgRNAs (green vertical lines) targeting the non-template (AS+28NT, AS+112NT) and template (AS+93NT) strands with respect to antisense transcription. The position of the Northern blotting probe detecting the *GAL1* AS transcript is shown in purple. (**C**) A Northern blot showing the reduction in *GAL1* antisense transcript (black arrowhead) in the strain expressing sgRNA AS+112NT/dCas9 relative to the no sgRNA control (*GAL1: ADH1*t *snR52::URA3* with dCas9). sgRNAs AS+28NT and AS+93T do not alter *GAL1* AS levels. Samples were taken from cells grown in glucose (t = 0) and at the indicated times after transfer to galactose-containing media (min). Positions of the rRNA are indicated by the short horizontal lines. Ethidium-bromide-stained rRNA is used as loading control. (**D**) Quantification of Northern blotting for the *GAL1* AS transcript at t = 0 in the control no sgRNA strain and the strain with sgRNA AS+112NT. N = 6, errors are SEM, *p=0.004. See *Source data 1*, tabs 10–12. (**E**) A map of the *HMS2* locus showing the *HMS2* AS transcript, *SUT650* (black arrow, transcript B) and positions of the three sgRNAs targeting *SUT650* (green vertical lines). The position of the Northern blotting probe to detect *SUT650* (H1 AS) is shown by the purple line. (**F**) A Northern blot probed with *HMS2* antisense probe H1 (see schematic in (**E**)) showing the level of *SUT650* (black arrowhead, transcript B) in the no sgRNA control (*snR52::URA3* with dCas9) strain and strains expressing the indicated antisense-targeting sgRNAs. Deletion of *XRN1* allows detection of a shorter antisense transcript ($B^t$) in the strain expressing AS+243NT. Positions of the rRNA are indicated by the short horizontal lines. *Represents cross-hybridisation with the 25S rRNA. A blot probed for the 18S rRNA is used as loading control. (**G**) Quantification of the level of *SUT650* (transcript B) reduction in the strains expressing each of the three antisense sgRNAs relative to the control no sgRNA strain. N = 4–8, errors represent SEM, *p<0.05. Click-linked and full-length synthetic sgRNA AS+243NT templates behave similarly. See *Source data 1*, tabs 5–9. (**H**) Northern blots with a series of *HMS2* antisense-specific probes. A new shorter antisense transcript ($B^t$) can be detected upon *XRN1* deletion in the strains expressing sgRNA AS+243NT/dCas9. Two clones of the CRISPRi strains produced using clicked (C) or full-length synthetic (FLS) DNA oligos for strain construction are shown. Positions of the antisense-specific probes (purple) and the site of sgRNA AS+243NT/dCas9 binding (green vertical line) are shown in the schematic. Positions of the rRNA on the Northern blot are indicated by the short horizontal lines. *Represents cross-hybridisation with the 25S rRNA. Ethidium-bromide-stained rRNA is used as loading control.

DOI: https://doi.org/10.7554/eLife.29878.004

The following figure supplements are available for figure 2:

**Figure supplement 1.** TIF-seq data showing the heterogeneity in *SUT650* transcript start site.

DOI: https://doi.org/10.7554/eLife.29878.005

**Figure supplement 2.** dCas9 is only detected at the *HMS2* AS+243NT sgRNA binding site.

DOI: https://doi.org/10.7554/eLife.29878.006

**Figure supplement 3.** Nucleosome occupancy over the *HMS2* sgRNA target sites does not anti-correlate with level of repression.

DOI: https://doi.org/10.7554/eLife.29878.007

(*Figure 3A*, t = 0). To support a strand-specific transcriptional block of antisense transcription, *GAL1* sense transcript polyA site usage was unaffected by sgRNA AS+112NT/dCas9 (*Figure 3C*). Furthermore, *XRN1* deletion in this strain also did not affect polyA site usage, ruling out a partial double-stranded transcriptional block and subsequent destabilisation of the resulting truncated sense transcript (*Figure 3C*).

## CRISPRi-induced *SUT650* repression truncates the *HMS2* sense transcript

Previous work shows that reducing *SUT650* transcription increases *HMS2* sense transcript levels (*Nguyen et al., 2014*). However, blocking *SUT650* by AS+243NT/dCas9 did not similarly increase *HMS2* sense levels (*Figure 3D*, lanes 1 and 3). To our surprise, in addition to the full-length *HMS2* sense transcript (A), we detected a considerably shortened *HMS2* sense transcript ($A^t$). Since the combined levels of the truncated and full-length transcripts are similar to those in the strain not expressing an sgRNA, independent of the presence of *XRN1* (*Figure 3D*, lanes 1 and 3 or 2 and 6), we hypothesised that the AS+243NT/dCas9 complex bound within the *HMS2* coding region may be causing partial premature sense transcription termination, leading to transcript $A^t$. Thus, we mapped the 3' end of *HMS2* S by Northern blotting using a series of strand-specific probes across the locus (*Figure 3E*). Both transcripts A and $A^t$ could clearly be detected using the probes upstream of the AS+243NT/dCas9 binding site (probes H1, H2, H3), but the truncated transcript was undetectable using probe H4 downstream of this site. Thus, the transcriptional block caused by AS+243NT/dCas9 is not strand-specific and re-defines the transcriptional landscape at the *HMS2* locus by creating a chromatin environment that is suitable for both transcription initiation and termination. To see if we could more directly target and block *HMS2* sense transcripts, we designed three sgRNAs to bind the non-template strand adjacent to PAM sites near the *HMS2* sense TSS (S+77NT, S+106NT, S+179NT) and monitored their effects on both *HMS2* sense and antisense transcripts (*Figure 3—*

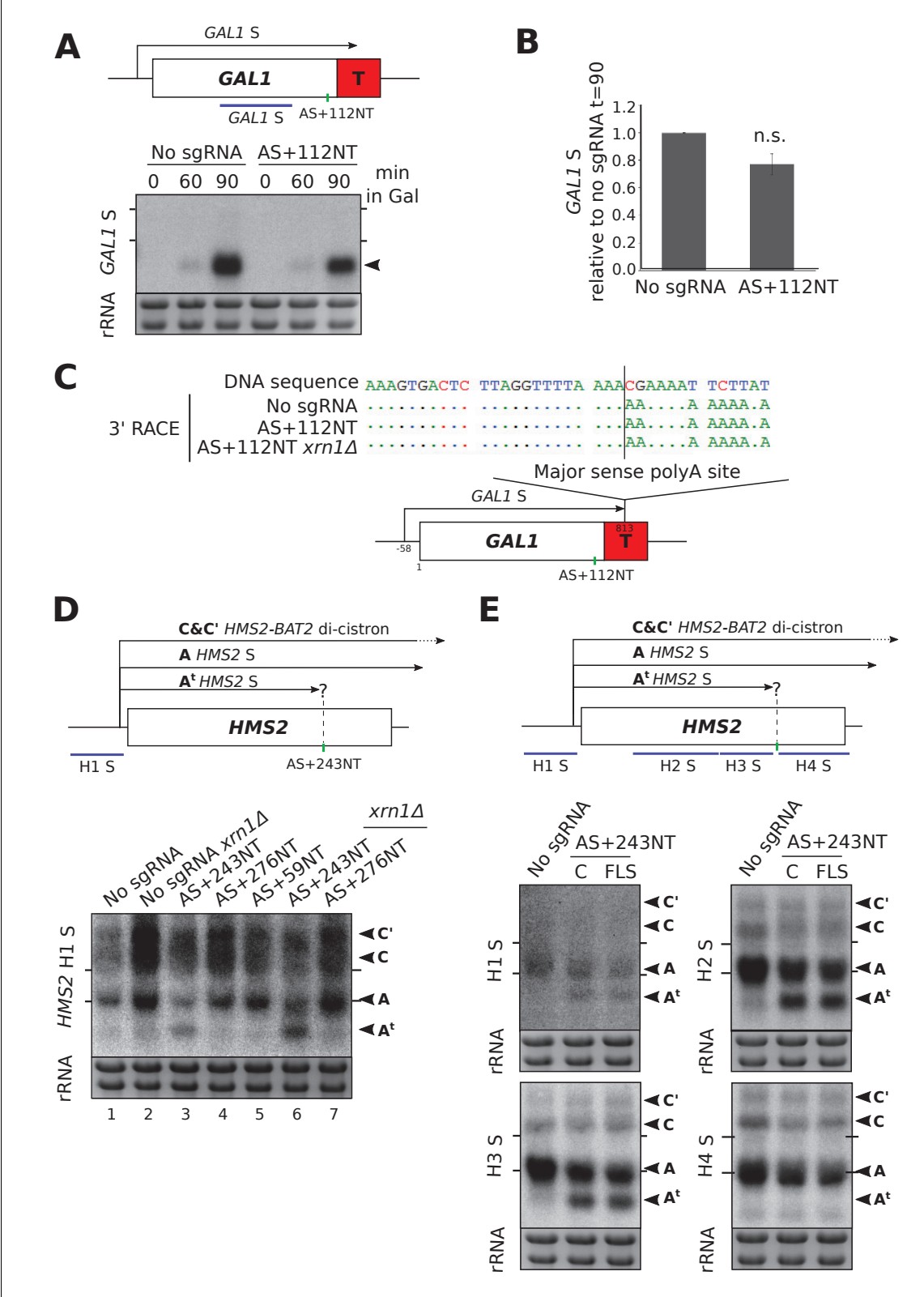

**Figure 3.** *HMS2* and *GAL1* sense transcripts are differently affected by blocking of their respective antisense transcripts. (**A**) A Northern blot probed for the *GAL1* sense transcript (black arrow) in the control strain with no sgRNA (*GAL1 ADH1*t *snR52::URA3* with dCas9) and in the strain expressing sgRNA AS+112NT/dCas9. Samples were taken from cells grown in glucose (t = 0) and at the indicated times after transfer to galactose-containing media (min). Ethidium-bromide-stained rRNA is used as a loading control. (**B**) Quantification of the Northern blot shown in (**A**) at t = 90. N = 4, error bars represent

*Figure 3 continued on next page*

*Figure 3 continued*

SEM, change is statistically non-significant (n.s.) p=0.062. See *Source data 1*, tabs 10–12. (C) A schematic showing the results of *GAL1* sense transcript 3′ end mapping by RACE in the strains indicated. All distances are shown relative to the sense TSS. Transcript cleavage and polyadenylation occurs beyond the position of the *GAL1* AS-blocking AS+112NT sgRNA shown (green vertical line). (D) A Northern blot probed with *HMS2* sense probe H1 (purple) showing the level of *HMS2* sense (black arrowhead, transcript A) in the no sgRNA control (*snR52::URA3* with dCas9) strain and strains expressing the indicated antisense-targeting sgRNAs. A truncated sense transcript (Aᵗ) was also detected in the strain expressing AS+243NT. C and C′ are read-through *HMS2-BAT2* transcripts (see [*Nguyen et al., 2014*]). Positions of the rRNA are indicated by the short horizontal lines. Ethidium-bromide-stained rRNA is used as loading control. (E) Northern blots with a series of *HMS2* sense-specific probes detecting the regions indicated in the schematic. The position of the sgRNA AS+243NT/dCas9 binding site is shown (green vertical line). Positions of the rRNA are indicated by the short horizontal lines. Ethidium-bromide-stained rRNA is used as loading control.
DOI: https://doi.org/10.7554/eLife.29878.008
The following figure supplement is available for figure 3:

**Figure supplement 1.** *HMS2* sense-targeting sgRNAs/dCas9 are not effective at repressing *HMS2* sense and do not alter antisense transcript levels.
DOI: https://doi.org/10.7554/eLife.29878.009

*figure supplement 1*). However, these sgRNAs gave no CRISPRi repression as indicated by the unchanged transcript levels and integrity.

## CRISPRi-induced antisense repression at *HMS2* and *GAL1* is not as efficient as previous methods

We compared the repression efficiency of CRISPRi with our previously used genetic interventions. At *GAL1*, mutation of a TATA-like sequence within the *ADH1* terminator greatly reduced the level of the *GAL1* antisense transcript (*Figure 4A,B*) (*Murray et al., 2015*). Lowered *GAL1* AS transcript levels could result from either reduced transcription or transcript destabilisation (or a combination of the two). To differentiate between these possibilities, we deleted the major cytoplasmic 5′−3′ exonuclease *XRN1*. Since *XRN1* deletion in the TATA mutant strain only slightly increased *GAL1* AS levels, this suggests that most repression was at the level of transcription rather than altering transcript stability (*Figure 4B,C*). By contrast, *XRN1* deletion in the strain expressing AS+112NT/dCas9 did result in some upregulation of *GAL1* AS (*Figure 4B,C*), indicating that antisense transcript repression was not as great in this strain. Unlike at *HMS2* (*Figure 2F,G*), this stabilised *GAL1* AS transcript was the same size as in the control and so likely represents stabilisation of the transcripts escaping repression rather than novel transcripts.

Next we compared the efficacy of AS+243NT/dCas9 with previous experiments to ablate *SUT650*, where we replaced the entire *HMS2* coding region, including the antisense TSS with the *URA3* coding region (*HMS2:URA3*) (*Nguyen et al., 2014*). This successful block of *SUT650* transcription increased levels of the *HMS2* sense transcript, the di-cistronic *HMS2-BAT2* transcript and consequently decreased levels of the downstream gene, *BAT2*, due to transcriptional interference (*Figure 4D–F*). Whilst CRISPRi resulted in a similar decrease in *SUT650* as *HMS2:URA3*, neither the *HMS2*, *BAT2* nor *HMS2-BAT2* di-cistronic transcripts were altered (*Figure 4F*). The discovery that whilst AS+243NT/dCas9 represses *SUT650*, a new unstable *HMS2* antisense transcript is induced and the sense transcript is prematurely terminated could explain why blocking *SUT650* using CRISPRi and the *URA3* gene body replacement strategies did not give the same results (*Figure 4F*). Thus, CRISPRi is not as effective as a genetic mutation in reducing levels of either the *GAL1* or *HMS2* AS transcripts.

## Concluding remarks

Although CRISPRi has been used to strand-specifically repress antisense transcription at *GAL10* (*Lenstra et al., 2015*) and *GAL1* (this work), here we demonstrate that this is not true at *HMS2*. This may reflect the vastly different transcription levels for the galactose-inducible genes (high) compared to *HMS2* (low). Furthermore, a study in human K562 cells found that CRISPRi-induced repression did not correlate with which strand was targeted at 49 genes, although the mechanism behind this observation was not investigated (*Gilbert et al., 2014*).

This work extends our previous hypothesis (*Nguyen et al., 2014*), allowing us to propose that, rather than acting as a roadblock, the binding of the sgRNA/dCas9 complex at *HMS2* creates a new chromatin environment, either directly or indirectly via the blocking of antisense transcription, which

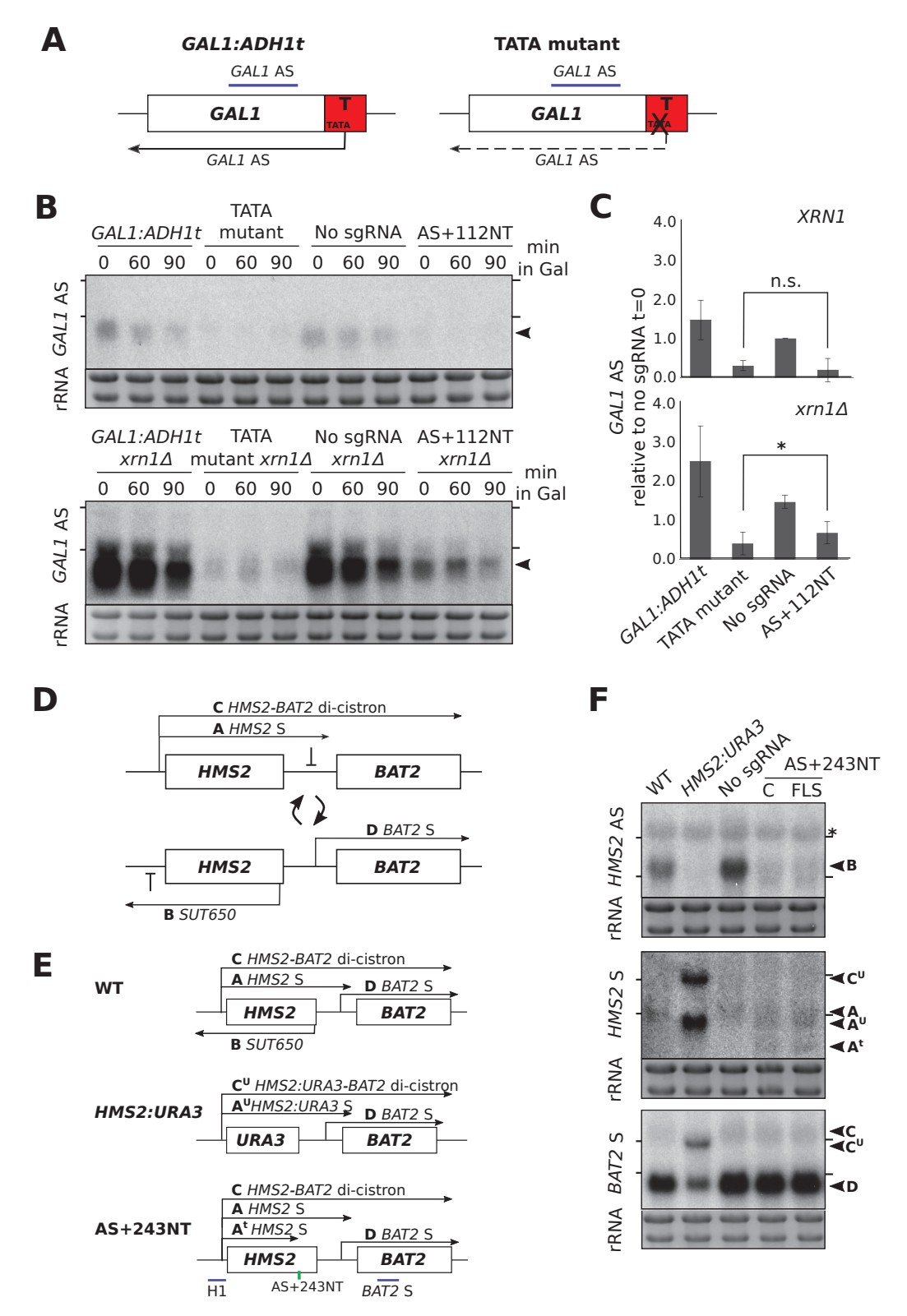

**Figure 4.** CRISPRi block of *SUT650* and *GAL1* AS is not as efficient as previous genetic interventions. (**A**) A schematic showing the previously used genetic intervention to repress the *GAL1* AS transcript (*Murray et al., 2015*). In the TATA mutant construct, a TATA box within the engineered *ADH1* terminator (T) is scrambled and the level of *GAL1* antisense transcript is reduced. (**B**) A Northern blot probed for the *GAL1* antisense transcript (black arrowhead) comparing the two methods of reducing antisense at this locus in the presence (top blot) or absence (bottom blot) of *XRN1*. All samples

*Figure 4 continued on next page*

*Figure 4 continued*

were run on the same gel and the blots were exposed to film for the same time. Samples were taken from cells grown in glucose (t = 0) and at the indicated times after transfer to galactose-containing media (min). Positions of the rRNA are indicated by the short horizontal lines. Ethidium-bromide-stained rRNA is used as loading control. (**C**) Quantification of the Northern blot in (**B**) at t = 0. N = 2, errors represent the SEM, *p=0.026, n.s. non-significant. See *Source data 1*, tab 12. (**D**) A schematic of the *HMS2-BAT2* locus, showing the *HMS2* sense and antisense transcripts (**A and B**), the *HMS2-BAT2* di-cistronic transcripts (**C and C'**) and the *BAT2* sense transcript (**D**). Cells switch between sense- and antisense-dominant states in which the indicated transcripts are produced (*Nguyen et al., 2014*). (**E**) The *HMS2-BAT2* locus and associated transcripts in the wild-type (WT) strain and strains that repress the *SUT650* transcript. In the *HMS2:URA3* strain, the entire coding region of *HMS2* is replaced by that of *URA3*. (**F**) Northern blots comparing the efficiency of *SUT650* reduction by replacement of the *HMS2* coding region with that of *URA3* versus CRISPRi with sgRNA AS+243NT and the relative effects of these two strategies on the *HMS2* sense and *BAT2* transcripts shown in (**E**). The WT (BY4741) and *HMS2.URA3* strains have been transformed with plasmid pRS315 to allow for growth on complete synthetic media lacking leucine and thus comparisons with the CRISPRi strains. Positions of the rRNA are indicated by the short horizontal lines. *Represents cross-hybridisation with the 25S rRNA. Ethidium-bromide-stained rRNA is used as loading control.

DOI: https://doi.org/10.7554/eLife.29878.010

is permissible for transcription initiation and termination. Thus, new transcription units are generated that result in novel sense and antisense transcripts of varying stabilities that are therefore not always detectable. This redefinition of the transcriptional landscape highlights the levels of controls and locus understanding needed before results from using CRISPRi can be interpreted.

# Materials and methods

## Key resource table

| Reagent type (species) or resource | Designation | Source or reference | Identifiers | Additional information |
|---|---|---|---|---|
| Gene (*Sacccharomyces cerevisiae*) | *HMS2* | | YJR147W; SGD ID S000003908; RRID:SCR_003093; SGD, RRID:SCR_004694 | |
| Gene (*S. cerevisiae*) | *GAL1* | | YBR020W; SGD ID S000000224; RRID:SCR_003093; SGD, RRID:SCR_004694 | |
| Gene (*S. cerevisiae*) | *BAT2* | | YJR148W; SGD ID S000003909 RRID:SCR_003093; SGD, RRID:SCR_004694 | |
| Gene (*S. cerevisiae*) | *snR52* | | SGD ID S000006443; RRID:SCR_003093; SGD, RRID:SCR_004694 | |
| Strain, strain background (*S. cerevisiae*) | see *Supplementary file 1C* | see *Supplementary file 1C* | *Saccharomyces cerevisiae*, NCBITaxon:1247190; SGD, RRID:SCR_004694 | see *Supplementary file 1C* |
| Genetic reagent (*S. cerevisiae*) | see *Supplementary file 1C* | see *Supplementary file 1C* | see *Supplementary file 1C* | see *Supplementary file 1C* |
| Antibody | Tubulin (rat monoclonal YL1/2) | Abcam | Abcam ab6160 RRID:AB_305328 | (1:3333) |
| Antibody | Cas9 (rabbit polyclonal) | Diagenode | C15310258 RRID:AB_2715516 | (1:5000 Western blot); 5 μl/ChIP |
| Recombinant DNA reagent | see *Supplementary file 1A* | Eurofins Genomics | see *Supplementary file 1A* | see *Supplementary file 1A* |
| Recombinant DNA reagent | Plasmid with p*TDH*-dCas9 | (*Qi et al., 2013*) Addgene | Addgene #46920 RRID:SCR:002037 | PMID:23849981 |

*Continued on next page*

*Continued*

| Reagent type (species) or resource | Designation | Source or reference | Identifiers | Additional information |
|---|---|---|---|---|
| Sequence-based reagent | single-stranded DNA templates for variable and constant sgRNA regions | This study | see *Supplementary file 1A* | synthesised by phosphoramidite oligonucleotide synthesis with 3'-alkyne or 5'-azido modifications (*El-Sagheer et al., 2011*) |
| Sequence-based reagent | see *Supplementary files 1A,B,D,E,F* | Eurofins Genomics | see *Supplementary files 1A,B,D,E,F* | see *Supplementary files 1A,B,D,E,F* |
| Commercial assay or kit | QIAquick PCR Purification Kit | Qiagen; RRID:SCR_008539 | Cat# 28106 | |
| Chemical compound, drug | SYBR green qPCR master mix | Bioline | Cat# QT605-20 | |
| Software, algorithm | Off-spotter sgRNA algorithm | https://cm.jefferson.edu/Off-Spotter// | | |
| Software, algorithm | Rotor-Gene Q Series Software | https://www.qiagen.com/us/resources/ | | |
| Software, algorithm | Fiji/Image J | https://imagej.net/Fiji | ImageJ, RRID:SCR_003070 | |

## sgRNA design

PAM sequences within 300 bp of the TSS in question were identified and the 20 bp immediately adjacent to these were used to design the variable regions of the sgRNAs. These sequences were run through an off-spotter algorithm (https://cm.jefferson.edu/Off-Spotter//) to minimise any off-target effects (*Pliatsika and Rigoutsos, 2015*). The sequence for the engineered constant region and *SUP4* terminator were taken from (*DiCarlo et al., 2013*; *Mali et al., 2013*). The *HMS2* sgRNA templates were synthesised as two DNA oligonucleotides to be joined by click chemistry, with the majority of the constant region on one oligonucleotide that could be used for all reactions. To increase the efficiency of the click reaction, the click linkage was designed between a dCpT dinucleotide within the Cas9 handle region. The sequences for the *HMS2* and *GAL1* sgRNA templates are shown in *Supplementary file 1A*. Full-length sgRNA templates were also synthesised for the *GAL1* sgRNAs and as a control for the *HMS2* click chemistry constructs (Eurofins Genomics, Germany).

## Click chemistry

Single-stranded DNA oligonucleotides complementary to the constant and variable sgRNA regions were synthesised by standard phosphoramidite oligonucleotide synthesis with 3'-alkyne or 5'-azido modifications respectively (*El-Sagheer et al., 2011*) (*Supplementary file 1A*). The click chemistry reaction was performed with a $Cu^1$ catalyst and 24–30 nt splint DNAs (*El-Sagheer et al., 2011*). PCR was performed for second strand synthesis, amplification and incorporation of sequences homologous to the site of insertion in *S. cerevisiae*. PCR efficiencies were obtained using qPCR performed three times on a Corbett 6000 Rotorgene with four serial 10-fold dilutions in triplicate. Analysis was performed using Rotor-Gene Q Series Software (Qiagen, Hilden, Germany).

## Strain construction

dCas9, without fusion to a transcriptional repressor domain, was expressed from a plasmid (# 46920 Addgene [*Qi et al., 2013*]) under the control of the *TDH3* promoter. The DNA templates for sgRNA production were integrated in place of an exogenous *URA3* cassette immediately downstream of the splice site in the endogenous *snR52* locus using homologous recombination followed by 5-FOA selection. Correct insertion was confirmed by genomic DNA Sanger sequencing. *SNR52* is a Pol III-transcribed C/D box small nucleolar RNA (snoRNA) gene and so does not undergo extensive post-transcriptional processing such as capping and polyadenylation, and transcripts from this locus are retained in the nucleus. Additionally, the locus contains a self-splicing site that produces a mature transcript without additional machinery, allowing precise production of a mature sgRNA without any unwanted extensions. Primer sequences for strain construction are listed in *Supplementary file 1B*.

## Yeast growth

The strains used in this study are listed in *Supplementary file 1C*. Strains were grown to mid-log at 30°C in complete synthetic media lacking leucine (for dCas9 plasmid selection). For experiments studying the *GAL1* locus, yeast were grown to mid-log in rich media (YP 2% D/YP 2% Gal) so that the *GAL1* induction kinetics were similar to what we had observed previously (*Murray et al., 2012*; *2015*). dCas9 expression was unaffected by the temporary absence of plasmid selection (*Figure 1—figure supplement 1*).

## Yeast growth controls

Assessment of growth in liquid culture (CSM-leucine) was achieved using a Bioscreen spectrophotometer that automatically measures the optical density of cultures at 30°C every 20 min for 24 hr. Doubling times were extracted from the gradient of the curves during logarithmic growth.

## Northern blotting

Northern blotting was performed as before (*Murray et al., 2012*; *Nguyen et al., 2014*) using asymmetric PCR or in vitro transcription with T7 RNA polymerase to generate the radioactive strand-specific probes for *GAL1* and *HMS2,* respectively. Primers for these reactions are listed in *Supplementary file 1D*. Northern blots were quantified using Fiji/ImageJ (*Schindelin et al., 2012*), and images were acquired using a FLA 7000 phosphorimager (GE Healthcare). Images in the figures are scans of exposures to X-ray film.

## Western blotting

dCas9 expression was confirmed after each experiment using an anti-Cas9 antibody (Diagenode C15310258) at 1:5000 dilution and anti-Tubulin (Abcam ab6160) at 1:3333.

## Chromatin immunoprecipitation

Yeast cells were fixed with 1% formaldehyde (30 min, 22°C) followed by the addition of glycine to 125 mM for 5 min. Cell pellets were collected by centrifugation (3000 rpm, 5 min) before washing twice with 10 ml cold PBS. Cells were re-suspended in cold FA-150 buffer (10 mM HEPES pH 7.9, 150 mM NaCl, 0.1% SDS, 0.1% sodium deoxycholate, 1% Triton X-100) and broken using 1-ml glass beads on a MagnaLyser (Roche) at ($2 \times 1$ min, 6000 rpm, 4°C). Fixed chromatin was sheared by sonication using a biorupter (Diagenode, 30 min, 30 s on, 30 s off, high setting), cleared by centrifugation (10,000 rpm, 15 min, 4°C) and incubated with 5 µl anti-Cas9 antibody (Diagenode C15310258) and 30 µl protein A-dynabeads pre-blocked with 200 µg/ml yeast tRNA, 200 µg/ml BSA and 200 µg/ml glycogen (15–20 hr, 4°C). Beads and attached chromatin were collected using a magnetic rack and washed with TSE-150 buffer (20 mM Tris-Cl pH 8.0, 150 mM NaCl, 2 mM EDTA, 0.1% SDS, 1% Triton X-100) for 3 min, TSE-500 buffer (20 mM Tris-Cl pH 8.0, 500 mM NaCl, 2 mM EDTA, 0.1% SDS, 1% Triton X-100) for 3 min, LiCl buffer (0.25 M LiCl, 10 mM Tris-Cl pH 8.0, 1 mM EDTA, 1% dioxycholate, 1 % NP-40) for 15 min and twice with TE, all at 22°C. After washing, chromatin was eluted from the beads for 30 min at 65°C with elution buffer (0.1 M NaHCO3, 1% SDS). Addition of 350 mM NaCl and incubation for 3 hr at 65°C reversed the cross-links before treatment of samples with RNase A for 1 hr at 37°C and proteinase K overnight at 65°C. DNA was purified using a PCR-purification kit (Qiagen) and eluted in 400 µl 1 mM Tris-Cl pH 8.0. Input DNA was diluted accordingly. Real-time quantitative PCR (qPCR) was carried out using a Corbett 6000 Rotorgene and Sybr green qPCR master mix (Bioline). Data ([IP - no antibody control]/input) were expressed as a percentage of the input (relative to sgRNA +276 NT). Real-time qPCR primers are listed in *Supplementary file 1E*.

## 3′ RACE mapping

Mapping of the 3′ end of the *GAL1* sense transcript was performed as (*Nguyen et al., 2014*). Primers are listed in *Supplementary file 1F*.

## Acknowledgements

The authors thank Jack Feltham for assistance with sgRNA template construction.

# Additional information

## Competing interests

Jane Mellor: Holds stock in Oxford BioDynamics Ltd., Chronos Therapeutics Ltd., and Sibelius Ltd. but these holdings present no conflict of interest with work in this article. The other authors declare that no competing interests exist.

## Funding

| Funder | Grant reference number | Author |
|--------|------------------------|--------|
| Biotechnology and Biological Sciences Research Council | BB/J001694/2 | Tom Brown Jane Mellor |
| Wellcome | 209897/Z/17/Z | Anna R Lamstaes |

The funders had no role in study design, data collection and interpretation, or the decision to submit the work for publication.

## Author contributions

Françoise S Howe, Conceptualization, Data curation, Formal analysis, Investigation, Methodology, Writing—original draft, Writing—review and editing; Andrew Russell, Anna R Lamstaes, Anitha Nair, Investigation; Afaf El-Sagheer, Conceptualization, Resources, Investigation, Methodology; Tom Brown, Conceptualization, Resources, Formal analysis, Supervision, Funding acquisition, Methodology, Writing—original draft, Project administration, Writing—review and editing; Jane Mellor, Conceptualization, Resources, Formal analysis, Supervision, Funding acquisition, Writing—original draft, Project administration, Writing—review and editing

## Author ORCIDs

Françoise S Howe  http://orcid.org/0000-0002-6455-2475
Afaf El-Sagheer  http://orcid.org/0000-0001-8706-1292
Tom Brown  http://orcid.org/0000-0002-6538-3036
Jane Mellor  http://orcid.org/0000-0002-5196-3734

## Decision letter and Author response

Decision letter https://doi.org/10.7554/eLife.29878.014
Author response https://doi.org/10.7554/eLife.29878.015

# Additional files

## Supplementary files

• Source data 1. Numerical data for all graphs (referenced in Figure Legends) Tab 1. Notes about yeast strain names Tab 2. Data supporting *Figure 1C* Tab 3. Data supporting *Figure 1D* Tab 4. Analysis of data in Tab 3 for *Figure 1D* Tab 5. Quantitation of all Northern blots for CRISPRi of *HMS2* AS transcripts supporting *Figure 2–4* Tab 6. Quantitation of all Northern blots for CRISPRi of *HMS2* S transcripts experiments 1 and 2 supporting *Figure 3—figure supplement 1*. Tab 7. Quantitation of all Northern blots for CRISPRi of *HMS2* S transcripts experiments 3 and 4 supporting *Figure 3—figure supplement 1*. Tab 8. Data analysis of comparison of CRISPRi versus genetic intervention (*URA3*) for HMS2 supporting *Figure 4D–F* Tab 9. *HMS2* sliding probes supporting *Figure 3E* and 4F Tab 10. *GAL1* AS CRISPRi data supporting *Figure 3A–B* Tab 11. All data for *GAL1* CRISPRi supporting *Figure 2*-4 Tab 12. Data analysis of comparison of CRISPRi versus genetic intervention (TATA mutant) for *GAL1* supporting *Figure 4B–C* Tab 13. Normalised dCas9 ChIP data at *HMS2* supporting *Figure 2—figure supplement 2*. Tab 14. Combined dCas9 ChIP data at *HMS2* for replicates 1–3 supporting *Figure 2—figure supplement 2*. Tab 15. Raw Ct data for dCas9 ChIP at *HMS2* replicate one supporting *Figure 2—figure supplement 2*. Tab 16. Raw Ct data for dCas9 ChIP at *HMS2* replicate two supporting *Figure 2—figure supplement 2*. Tab 17. Raw Ct data for dCas9 ChIP at *HMS2*

replicate three supporting *Figure 2—figure supplement 2*. All raw data can be found at Mendeley Data: http://dx.doi.org/10.17632/p2652vcdz8.2.

DOI: https://doi.org/10.7554/eLife.29878.011

• Supplementary file 1. Supplementary Tables (referenced in the Materials and methods). (A) Templates for sgRNAs (B) Primers for strain construction (C) *Saccharomyces cerevisiae* strains used in this study (D) Primers for construction of Northern blot probe templates (E) Primers for real-time qPCR at *HMS2* (F) Primers for 3′ RACE mapping of the *GAL1* sense transcript.

DOI: https://doi.org/10.7554/eLife.29878.012

• Transparent reporting form

DOI: https://doi.org/10.7554/eLife.29878.013

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
