## [Decision Letter]

Thank you for submitting your article "CRISPRi is not strand-specific and redefines the transcriptional landscape" for consideration by *eLife*. Your article has been reviewed by three peer reviewers, one of whom is a member of our Board of Reviewing Editors, and the evaluation has been overseen by James Manley as the Senior Editor. We have attached the reviewers' comments for you below.

As you will see, the reviewers find your study of great interest to the community, but have also suggested that the paper is technical in nature and somewhat lacks mechanistic insight. We are willing to consider a revised version of your manuscript addressing the comments/concerns of the reviewers, specifically reviewer 2.

Reviewer #1:

In recent years, transcriptional interference by antisense RNA transcription has been recognized as a relatively frequent means to regulate mRNA expression. Tools enabling antisense transcription arrest without directly affecting sense transcription are required to study the mechanism mediating this effect. The recently reported CRISPRi system, using a modified, partially defective, Cas9 (dCas9) appeared as ideally suited for this since it has been shown to be able to trigger transcription termination in a site specific and, most importantly, strand specific manner, without altering the genomic sequence.

This short report shows that this strand specificity is actually not true at all loci. In addition, it describes an example in which the binding of the guide RNA actually induces spurious transcription initiation, suggesting that it markedly perturbs the local chromatin environment.

I think that the data are suitably convincing. Note, by the way, that this referee has made the similar (unpublished) observations that the termination by this system was not strand specific at yet another locus.

Although one could argue that the main results presented in this manuscript are negative results, I personally think that they are well worth being published to warn further investigators who would wish to use this system and a short report in *eLife* might seem appropriate for that.

The position of the 5'-end of SUT650 seems incorrectly assigned. Indeed, if one refers to the TIF-seq data from the original Pelechano paper, the main transcription start site for SUT650 (position 705244 on the Crick strand of chromosome X; this is also consistent with the data from Malabat et al., *eLife*, 2015) would be located about 60 nucleotides DOWNSTREAM of the AS-44NT sgRNA binding site (position 705184). Thus, as the other SUT650 sgRNA targeting sequences, AS-44NT would actually be located within the SUT650 transcribed region, downstream from its TSS and not upstream. It could thus not be considered as a "control" sgRNA. Could the authors verify this point and, if this is correct, modify the text, Figure 2 and Figure 2—figure supplement 1 accordingly? This would in no way modify the conclusions of the manuscript.

Reviewer #2:

In this paper by Howe et al., the authors assess the potential of CRISPRi for strand-specific transcriptional perturbation in budding yeast. They study two genes in depth – GAL1 and HMS2 – and compare the effectiveness of CRISPRi to previous approaches (for example deletion of cis-acting motifs such as the TATA box) in wt and Xrn1 deletion strains. Their primary conclusion is that CRISPRi works well in some cases and not in others and requires careful controls. For example, the authors show that the system does repress antisense transcription, but at one of the two studied loci (HMS2), an additional antisense transcript appears from a new initiation site, and the sense transcript is prematurely terminated. In addition, they present a clever tool based on click chemistry for quickly making gRNA. Although I don't find the main conclusion revelatory, I commend the authors for their thoroughness. Certainly, this message is one that bears reporting, although I am not quite convinced this result clears the bar for an *eLife* paper.

1) The authors show that CRISPRi represses the antisense transcript at a GAL1 model gene. They then check whether this repression has transcriptional effects on GAL1 sense transcription and they conclude there is no difference in galactose. However, previous work has shown that antisense at the GAL locus works mostly under repressive conditions. Have the authors repeated this experiment in glucose or raffinose?

2) The authors show that CRISPRi repression at the HMS2 locus results in a changed transcriptional landscape. Replacement of the HMS locus by URA does not have the same effect, and they suggest that CRISPRi binding causes these additional transcripts. However, it is unclear whether the new transcripts at HMS2 appear because of removal of the antisense, which in itself could change the chromatin landscape and thus the transcriptional landscape, or whether it arises because of binding of dCas9. The same authors have previously published that removal of the GAL10 ncRNA by genetic methods (point mutations of the transcription factor binding sites) also results in additional transcripts not visible in the wildtype (Murray et al., 2015, supplementary figure S5C). Given these findings, it remains unclear whether the new antisense transcript indeed results from the CRISPRi binding. Moreover, it is noteworthy that additional transcripts are also found in some genetic mutants. Control experiments are therefore required for both methods.

3) The Xrn1 sensitivity is a confounding factor, and I am not sure that strain should be the gold standard for CRISPRi not working. The whole point is that one can get blocking or perturbation in the absence of genetic manipulation. So, for example one concluding sentence "CRISPRi is not as effective as a genetic mutation in reducing levels of either the GAL1 or HMS2 AS transcripts" should end with "[…] in the Xrn1 deletion."

4) In a related point, Figure 4 is a critical part of the argument. The lower panel needs to have significance calculated between the TATA mutant and AS+112NT, since these bars look pretty similar to this referee. Moreover, the similarity between these measurements suggests that the lower panel in Figure 4 is not quite a representative image.

5) Lastly, given that CRISPRi does represses the antisense transcript in both loci tested in the study in a strand-specific manner, the title is a bit misleading.

In summary, despite technical issues, I nevertheless find their argument that CRISPRi can have locus-specific effects to be convincing. However, the manuscript is mostly of a technical nature. Most experiments described are control experiments that should be done anyways when using a new method. The observations that CRISPRi can result in different transcripts is good to know when designing such experiments, but there is no proposed mechanistic insight, nor new biology that is learned from this study.

Reviewer #3:

In this manuscript Howe and collaborators in the Mellor and Brown labs explore the molecular basis of CRISPRi transcriptional interference in the context of yeast anti-sense transcription. This reviewer agrees with the authors that understanding the basis of CRISPRi in anti-sense transcription (and non-coding RNA transcription in general) is an important area of research because it is extremely difficult to establish direct causality with regard to these particular RNAs using genetic perturbations.

In the first section of their manuscript, Howe et al., describe synthesis of sgRNA constructs using "click-chemistry" to couple sgRNA variable regions to the constant region. The author's then use knock-in yeast strains generated using these constructs to evaluate the effects of CRISPRi on anti-sense transcription at the GAL1 and HMS2 genes. The authors observe that contrary to behaving as a transcriptional road-block when targeted to the non-template strand, dCas9 CRISPRi complexes result in unpredictable changes in chromatin and transcription, resulting in pre-mature termination of sense transcripts and shifting anti-sense transcription initiation sites. Collectively, these results clearly demonstrate that CRISPRi effects are more complex than previously reported and likely depend on locus-specific chromatin context. These observations have broad implications with respect to design of CRISPRi strategies and argue a strong case for careful characterization of transcript isoforms and promoter/termination sites before and after deploying CRISPRi. However, there are some fundamental issues with controls and some of the conclusions drawn that should be addressed before publication.

1) The authors observe that only a subset of sgRNAs targeting the non-template strand are capable of mediating repression, however, the reasons for this phenomenon are unclear. A simple explanation is that some sgRNAs are not capable of mediating stable dCAS9 interaction with chromatin. The author's should test this by performing Chromatin-Immunoprecipitation (ChIP) for dCas9 in their various sgRNA expressing lines to test if dCas9 chromatin occupancy correlates with repression. In my opinion this is an absolutely essential control for interpreting the authors' experiments.

2) This reviewer feels that increasing the sgRNA coverage across the sense/antisense transcripts in question would greatly strengthen the manuscript. In its current state, the authors test only 2 sgRNAs towards the GAL1-AS and SUT650 non-template strands, from which they obtain a single functional repressive sgRNA for each anti-sense transcript. As a strand specificity control they test 3 sgRNAs targeting the sense transcript HMS2 and show that none of these has an effect on sense or anti-sense. However, with a hit rate of ~1 in 2, this reviewer is not convinced by authors' claim that sgRNAs targeting the sense transcript have no effect on antisense transcription. Perhaps the coverage was simply too low to obtain a functional repressive sgRNA. The authors should perform high-density sgRNA tiling experiments (targeting both template and non-template) for either GAL1-AS or SUT650 (not necessary to do both loci). I feel this experiment will dramatically improve the manuscript and instill clarity into the context requirements for CRISPRi mediated repression.

---

## [Author Response]

Reviewer #1:In recent years, transcriptional interference by antisense RNA transcription has been recognized as a relatively frequent means to regulate mRNA expression. Tools enabling antisense transcription arrest without directly affecting sense transcription are required to study the mechanism mediating this effect. The recently reported CRISPRi system, using a modified, partially defective, Cas9 (dCas9) appeared as ideally suited for this since it has been shown to be able to trigger transcription termination in a site specific and, most importantly, strand specific manner, without altering the genomic sequence.This short report shows that this strand specificity is actually not true at all loci. In addition, it describes an example in which the binding of the guide RNA actually induces spurious transcription initiation, suggesting that it markedly perturbs the local chromatin environment.I think that the data are suitably convincing. Note, by the way, that this referee has made the similar (unpublished) observations that the termination by this system was not strand specific at yet another locus.Although one could argue that the main results presented in this manuscript are negative results, I personally think that they are well worth being published to warn further investigators who would wish to use this system and a short report in eLife might seem appropriate for that.

We thank the reviewer for recognising the importance of this work as a warning to other researchers.

The position of the 5'-end of SUT650 seems incorrectly assigned. Indeed, if one refers to the TIF-seq data from the original Pelechano paper, the main transcription start site for SUT650 (position 705244 on the Crick strand of chromosome X; this is also consistent with the data from Malabat et al., eLife, 2015) would be located about 60 nucleotides DOWNSTREAM of the AS-44NT sgRNA binding site (position 705184). Thus, as the other SUT650 sgRNA targeting sequences, AS-44NT would actually be located within the SUT650 transcribed region, downstream from its TSS and not upstream. It could thus not be considered as a "control" sgRNA. Could the authors verify this point and, if this is correct, modify the text, Figure 2 and Figure 2—figure supplement 1 accordingly? This would in no way modify the conclusions of the manuscript.

We thank the reviewer for pointing this out and apologise for our oversight. We note that there is a large heterogeneity in the position of the transcription start site for SUT650 and the one we had chosen previously (position 705149, published in Nguyen et al., 2014) is one of the many sites. We have now discussed the transcript heterogeneity in the main body of the text and included a supplementary figure with the Pelechano TIF-seq data to illustrate this (Figure 2—figure supplement 1). We have also renamed the HMS2 strains in the text and figures based on the new AS-targeting sgRNA positions relative to the major SUT650 TSS at position 705244 identified by this reviewer.

Reviewer #2:In this paper by Howe et al., the authors assess the potential of CRISPRi for strand-specific transcriptional perturbation in budding yeast. They study two genes in depth – GAL1 and HMS2 – and compare the effectiveness of CRISPRi to previous approaches (for example deletion of cis-acting motifs such as the TATA box) in wt and Xrn1 deletion strains. Their primary conclusion is that CRISPRi works well in some cases and not in others and requires careful controls. For example, the authors show that the system does repress antisense transcription, but at one of the two studied loci (HMS2), an additional antisense transcript appears from a new initiation site, and the sense transcript is prematurely terminated. In addition, they present a clever tool based on click chemistry for quickly making gRNA. Although I don't find the main conclusion revelatory, I commend the authors for their thoroughness. Certainly, this message is one that bears reporting, although I am not quite convinced this result clears the bar for an eLife paper.

We thank the reviewer for their support of the message we are trying to convey to the community. We feel that *eLife*, with its reputation for thoroughness and research integrity, would be an ideal journal for publishing this work.

1) The authors show that CRISPRi represses the antisense transcript at a GAL1 model gene. They then check whether this repression has transcriptional effects on GAL1 sense transcription and they conclude there is no difference in galactose. However, previous work has shown that antisense at the GAL locus works mostly under repressive conditions. Have the authors repeated this experiment in glucose or raffinose?

As presented in Figure 3, we observed no leaky expression of GAL1 sense transcript in glucose (t=0) in the absence of the antisense and the sense induction kinetics are similar to the control. We have included a sentence in the manuscript text to explain this.

2) The authors show that CRISPRi repression at the HMS2 locus results in a changed transcriptional landscape. Replacement of the HMS locus by URA does not have the same effect, and they suggest that CRISPRi binding causes these additional transcripts. However, it is unclear whether the new transcripts at HMS2 appear because of removal of the antisense, which in itself could change the chromatin landscape and thus the transcriptional landscape, or whether it arises because of binding of dCas9. The same authors have previously published that removal of the GAL10 ncRNA by genetic methods (point mutations of the transcription factor binding sites) also results in additional transcripts not visible in the wildtype (Murray et al., 2015, supplementary figure S5C). Given these findings, it remains unclear whether the new antisense transcript indeed results from the CRISPRi binding. Moreover, it is noteworthy that additional transcripts are also found in some genetic mutants. Control experiments are therefore required for both methods.

The point made here by this reviewer is a valid one – we are unable to infer whether it is the binding of sgRNA/dCas9 or the loss of the antisense that is causing the initiation of the new transcripts. It would be experimentally very difficult to show this and we have now clarified the two possible reasons for new transcripts in the text.

However, the main point is that, independent of what is causing the initiation of the new antisense transcript, in the CRISPRi strain, transcription is occurring in the antisense direction into the HMS2 sense promoter and so no conclusions could be made about the function of SUT650 in sense regulation in this strain. The fear is that some investigators may use the CRISPRi system without the proper controls and the current set of experiments is intended to highlight how important these controls are.

3) The Xrn1 sensitivity is a confounding factor, and I am not sure that strain should be the gold standard for CRISPRi not working. The whole point is that one can get blocking or perturbation in the absence of genetic manipulation. So, for example one concluding sentence "CRISPRi is not as effective as a genetic mutation in reducing levels of either the GAL1 or HMS2 AS transcripts" should end with "… in the Xrn1 deletion."

We apologise that we were not clear about the reason for performing the experiments in the XRN1 delete and have now clarified this in the text. We chose to use the deletion of XRN1 to stabilise any transcripts that may be being produced in the CRISPRi strains but then rapidly degraded so that, if the experiment is done in an XRN1^+^ background, the transcript could not be detected. This is important mechanistically because often it is the act of antisense transcription that may cause the effect on neighbouring/overlapping transcription units rather than the antisense transcript itself (e.g. Murray et al., 2015). In Figure 4, both the TATA mutant and AS+112NT have undetectable GAL1 AS in the top panel. However, upon XRN1 deletion, a little GAL1 AS is stabilised in the TATA mutant but much more is stabilised in the AS+112NT strain. We interpret this to indicate that there is more leaky but unstable GAL1 AS transcript in the AS+112NT strain than in the TATA mutant strain and thus it is true to say that CRISPRi is not as effective as a genetic mutation in reducing levels of GAL1 AS transcriptionin the presence or absence of XRN1. The same logic can also be applied to the new unstable Xrn1-sensitive HMS2 AS transcripts that appear in the AS+243NT (previously called AS+148NT) strain.

4) In a related point, Figure 4 is a critical part of the argument. The lower panel needs to have significance calculated between the TATA mutant and AS+112NT, since these bars look pretty similar to this referee. Moreover, the similarity between these measurements suggests that the lower panel in Figure 4 is not quite a representative image.

We have found it very difficult to quantify blots with very low signal levels so we apologise that the image did not appear to be representative of the accompanying quantification. We have now included a more rigorous background subtraction step for the GAL1 AS Northerns and hope that the reviewer is satisfied with the new Figure 4. We have also included significance values here. The same alteration in the quantification of GAL1 AS transcript levels has been repeated for the other GAL1 AS blots and Figure 2 has therefore been remade accordingly.

5) Lastly, given that CRISPRi does represses the antisense transcript in both loci tested in the study in a strand-specific manner, the title is a bit misleading.

We have shown that CRISPRi only works strand-specifically at GAL1, because at HMS2, both the sense and the antisense transcripts are blocked. However, we agree that the original title perhaps didn’t reflect this fully so have now changed it to include the words ‘at all loci’: ‘CRISPRi is not strand-specific at all loci and redefines the transcriptional landscape’.

In summary, despite technical issues, I nevertheless find their argument that CRISPRi can have locus-specific effects to be convincing. However, the manuscript is mostly of a technical nature. Most experiments described are control experiments that should be done anyways when using a new method. The observations that CRISPRi can result in different transcripts is good to know when designing such experiments, but there is no proposed mechanistic insight, nor new biology that is learned from this study.

We agree that most of the experiments we have done should be performed as controls for any CRISPRi experiments but are aware that, especially if CRISPRi is being used for a large-scale screen, this probably won’t be done (and may not be a practical proposition). We wish to publish this work to highlight how important it is that these controls are done or perhaps to suggest that CRIPSRi may not always be a suitable technique if the correct controls cannot be performed.

We have also added mechanistic insight in the new Figure 2—figure supplement 2, where we have performed a ChIP experiment for Cas9 and show that dCas9 is present at the site of sgRNA binding for the sgRNA that successfully represses HMS2 AS transcript but not at those sgRNAs that do not.

Reviewer #3:In this manuscript Howe and collaborators in the Mellor and Brown labs explore the molecular basis of CRISPRi transcriptional interference in the context of yeast anti-sense transcription. This reviewer agrees with the authors that understanding the basis of CRISPRi in anti-sense transcription (and non-coding RNA transcription in general) is an important area of research because it is extremely difficult to establish direct causality with regard to these particular RNAs using genetic perturbations.In the first section of their manuscript, Howe et al., describe synthesis of sgRNA constructs using "click-chemistry" to couple sgRNA variable regions to the constant region. The author's then use knock-in yeast strains generated using these constructs to evaluate the effects of CRISPRi on anti-sense transcription at the GAL1 and HMS2 genes. The authors observe that contrary to behaving as a transcriptional road-block when targeted to the non-template strand, dCas9 CRISPRi complexes result in unpredictable changes in chromatin and transcription, resulting in pre-mature termination of sense transcripts and shifting anti-sense transcription initiation sites. Collectively, these results clearly demonstrate that CRISPRi effects are more complex than previously reported and likely depend on locus-specific chromatin context. These observations have broad implications with respect to design of CRISPRi strategies and argue a strong case for careful characterization of transcript isoforms and promoter/termination sites before and after deploying CRISPRi. However, there are some fundamental issues with controls and some of the conclusions drawn that should be addressed before publication.1) The authors observe that only a subset of sgRNAs targeting the non-template strand are capable of mediating repression, however, the reasons for this phenomenon are unclear. A simple explanation is that some sgRNAs are not capable of mediating stable dCAS9 interaction with chromatin. The author's should test this by performing Chromatin-Immunoprecipitation (ChIP) for dCas9 in their various sgRNA expressing lines to test if dCas9 chromatin occupancy correlates with repression. In my opinion this is an absolutely essential control for interpreting the authors' experiments.

We are not the first to observe that, as for CRISPR, some sgRNAs are not effective for CRISPRi e.g. Lenstra et al., 2015. We do not aim to address the question of why only some sgRNAs are repressive in this paper. However, we have now performed a ChIP experiment for (d)Cas9 (see Figure 2—figure supplement 2) to show that we can’t detect dCas9 binding at the sgRNA sites that gave us no repression but have a strong dCas9 signal at the binding site for sgRNA AS+243NT (previously AS+148NT). Thus we have added mechanistic detail to the phenomenon observed by us and others.

2) This reviewer feels that increasing the sgRNA coverage across the sense/antisense transcripts in question would greatly strengthen the manuscript. In its current state, the authors test only 2 sgRNAs towards the GAL1-AS and SUT650 non-template strands, from which they obtain a single functional repressive sgRNA for each anti-sense transcript. As a strand specificity control they test 3 sgRNAs targeting the sense transcript HMS2 and show that none of these has an effect on sense or anti-sense. However, with a hit rate of ~1 in 2, this reviewer is not convinced by authors' claim that sgRNAs targeting the sense transcript have no effect on antisense transcription. Perhaps the coverage was simply too low to obtain a functional repressive sgRNA. The authors should perform high-density sgRNA tiling experiments (targeting both template and non-template) for either GAL1-AS or SUT650 (not necessary to do both loci). I feel this experiment will dramatically improve the manuscript and instill clarity into the context requirements for CRISPRi mediated repression.

We do not claim that, if we were to find an sgRNA to block HMS2 sense, that this would not also affect the antisense transcript rather that the 3 sgRNAs we tested against HMS2 sense did not repress either the sense or antisense transcripts. We apologise if this was confused in the text and have clarified this section. We included the (negative) results from the 3 sense-targeting sgRNAs again to reiterate the fact that not all sgRNAs are effective and this is an important control to do. We included the HMS2 AS data from the sense sgRNA strains as a control for the sgRNAs also not working on the template strand.

As stated above, we are not aiming to investigate what makes some sgRNAs cause effective transcript repression and others not. We do not feel that testing more sgRNAs would significantly add to the conclusions of the paper, which were that a) CRIPSRi does not always repress transcription b) CRISPRi is not always strand-specific and c) CRISPRi can induce novel transcripts that would confound any analysis. It is also relevant that the highest CRISPRi activity with unfused dCas9 was observed using PAM sites near the relevant TSSs (Qi et al., 2013) as we have done here. Using PAM sites that are more distant from the TSS is unlikely to add any insight into the conclusions we have made here.